# An interactive meta-analysis of MRI biomarkers of myelin

**Matteo Mancini[1,2,3]\*, Agah Karakuzu[2], Julien Cohen-Adad[2,4], Mara Cercignani[1,5], Thomas E Nichols[6,7†], Nikola Stikov[2,8†]**

[1]Department of Neuroscience, Brighton and Sussex Medical School, University of Sussex, Brighton, United Kingdom; [2]NeuroPoly Lab, Polytechnique Montreal, Montreal, Canada; [3]CUBRIC, Cardiff University, Cardiff, United Kingdom; [4]Functional Neuroimaging Unit, CRIUGM, Université de Montréal, Montreal, Canada; [5]Neuroimaging Laboratory, Fondazione Santa Lucia, Rome, Italy; [6]Wellcome Centre for Integrative Neuroimaging (WIN FMRIB), University of Oxford, Oxford, United Kingdom; [7]Big Data Institute, University of Oxford, Oxford, United Kingdom; [8]Montreal Heart Institute, Université de Montréal, Montreal, Canada

**Abstract** Several MRI measures have been proposed as in vivo biomarkers of myelin, each with applications ranging from plasticity to pathology. Despite the availability of these myelin-sensitive modalities, specificity and sensitivity have been a matter of discussion. Debate about which MRI measure is the most suitable for quantifying myelin is still ongoing. In this study, we performed a systematic review of published quantitative validation studies to clarify how different these measures are when compared to the underlying histology. We analyzed the results from 43 studies applying meta-analysis tools, controlling for study sample size and using interactive visualization (https://neurolibre.github.io/myelin-meta-analysis). We report the overall estimates and the prediction intervals for the coefficient of determination and find that MT and relaxometry-based measures exhibit the highest correlations with myelin content. We also show which measures are, and which measures are not statistically different regarding their relationship with histology.

**\*For correspondence:**
ingmatteomancini@gmail.com

†These authors contributed equally to this work

**Competing interests:** The authors declare that no competing interests exist.

## Introduction

Myelin is a key component of the central nervous system. The myelin sheaths insulate axons with a triple effect: allowing fast electrical conduction, protecting the axon, and providing trophic support (*Nave and Werner, 2014*). The conduction velocity regulation has become an important research topic, with evidence of activity-dependent myelination as an additional mechanism of plasticity (*Fields, 2015*; *Sampaio-Baptista and Johansen-Berg, 2017*). Myelin is also relevant from a clinical perspective, given that demyelination is often observed in several neurological diseases such as multiple sclerosis (*Höftberger and Lassmann, 2018*).

Given this important role in pathology and plasticity, measuring myelin in vivo has been an ambitious goal for magnetic resonance imaging (MRI) for more than two decades (*MacKay et al., 1994*; *Rooney et al., 2007*; *Stanisz et al., 1999*). Even though the thickness of the myelin sheath is in the order of micrometres, well beyond the MRI spatial resolution, its presence influences several physical properties that can be probed with MRI, from longitudinal and transversal relaxation phenomena to water molecule diffusion processes.

However, being sensitive to myelin is not enough: to study how and why myelin content changes, it is necessary to define a specific biomarker. Interestingly, the quest for measuring myelin has evolved in parallel with an important paradigm shift in MRI research, where MRI data are no longer treated as just 'pictures', but as actual 3D distributions of quantitative measures. This perspective

has breathed new life into an important field of research, quantitative MRI (qMRI), that encompasses the study of how to measure the relevant electromagnetic properties that influence magnetic resonance phenomena in biological tissues (*Cercignani et al., 2018*; *Cohen-Adad and Wheeler-Kingshott, 2014*). From the very definition of qMRI, it is clear that its framework applies to any approach for non-invasive myelin quantification.

Similarly to other qMRI biomarkers, MRI-based myelin measurements are indirect, and might be affected by other microstructural features, making the relationship between these indices and myelination noisy. Assessing the accuracy of such measurements, and their sensitivity to change, is essential for their translation into clinical applications. Validation is therefore a fundamental aspect of their development (*Cohen-Adad, 2018*). The most common approach is based on acquiring MR data from in vivo or ex vivo tissue and then comparing those data with the related samples analyzed using histological techniques. Despite being the most realistic approach, this comparison involves several methodological choices, from the specific technique used as a reference to the quantitative measure used to describe the relationship between MRI and histology. So far, a long list of studies have looked at MRI-histology comparisons (*Cohen-Adad, 2018*; *Laule and Moore, 2018*; *MacKay and Laule, 2016*; *Petiet et al., 2019*), each of them focusing on a specific pathology and a few MRI measures.

Despite these numerous studies, there is still an ongoing debate on what MRI measure should be used to quantify myelin and as a consequence there is a constant methodological effort to propose new measures. This debate would benefit from a quantitative analysis of all the findings published so far, specifically addressing inter-study variations and prospects for future studies, something that is currently missing from the literature.

In this study, we systematically reviewed quantitative MRI-histology comparisons and we used meta-analysis tools to address the following question: how different are the modalities for myelin quantification in terms of their relationship with the underlying histology?

## Results

### Literature survey

The screening process is summarized in the flowcharts in *Figure 1* and *Appendix 1—figure 1*. The keywords as reported in the appendix returned 688 results on PubMed (last search on 03/06/2020). These results included 50 review articles. From the 50 review articles, six were selected as relevant for both the topics of myelin and related MRI-histology comparisons (*Cohen-Adad, 2018*; *Laule and Moore, 2018*; *Laule et al., 2007*; *MacKay and Laule, 2016*; *Petiet et al., 2019*; *Turner, 2019*). After the assessment, 58 original research studies were considered eligible, as shown in *Appendix 1—table 1* (in the appendix) and Figure S2. All the data collected are available in the supplementary materials (*Source data 1*).

In terms of specific modalities, the survey shows that the most common MRI approach compared with histology was diffusion-weighted imaging (used in 28 studies), followed by magnetization transfer (MT, 27 studies), T2 relaxometry (19 studies) and T1 relaxometry (10 studies). Only 20 studies considered more than one approach: among the others, 20 focused exclusively on diffusion, 12 on MT, and six on T2 relaxometry.

From these 58 studies, we then focused only on brain studies and we further excluded studies not reporting either the number of subjects or the number of ROIs per subject. We also excluded one single-subject study that relied on voxels as distinct samples, whereas the other studies in this review are based on ROIs (i.e. including more than one voxel). In the end, 43 suitable studies were identified for the subsequent analyses.

### Meta-analysis

To compare the studies of interest, we first organized them according to the MRI measure used. *Figure 2* and *Figure 3* (and also Figure S3-S4) show the $R^2$ values for the selected studies across measures: the highest values ($R^2 > 0.8$) are obtained mostly from MT measures, but they are associated with small sample sizes (with an average of 32 sample points). The studies with largest sample sizes are associated with $R^2$ values between 0.6 and 0.8 for MT and T2 relaxometry, but with lower values for T1 relaxometry and other approaches.

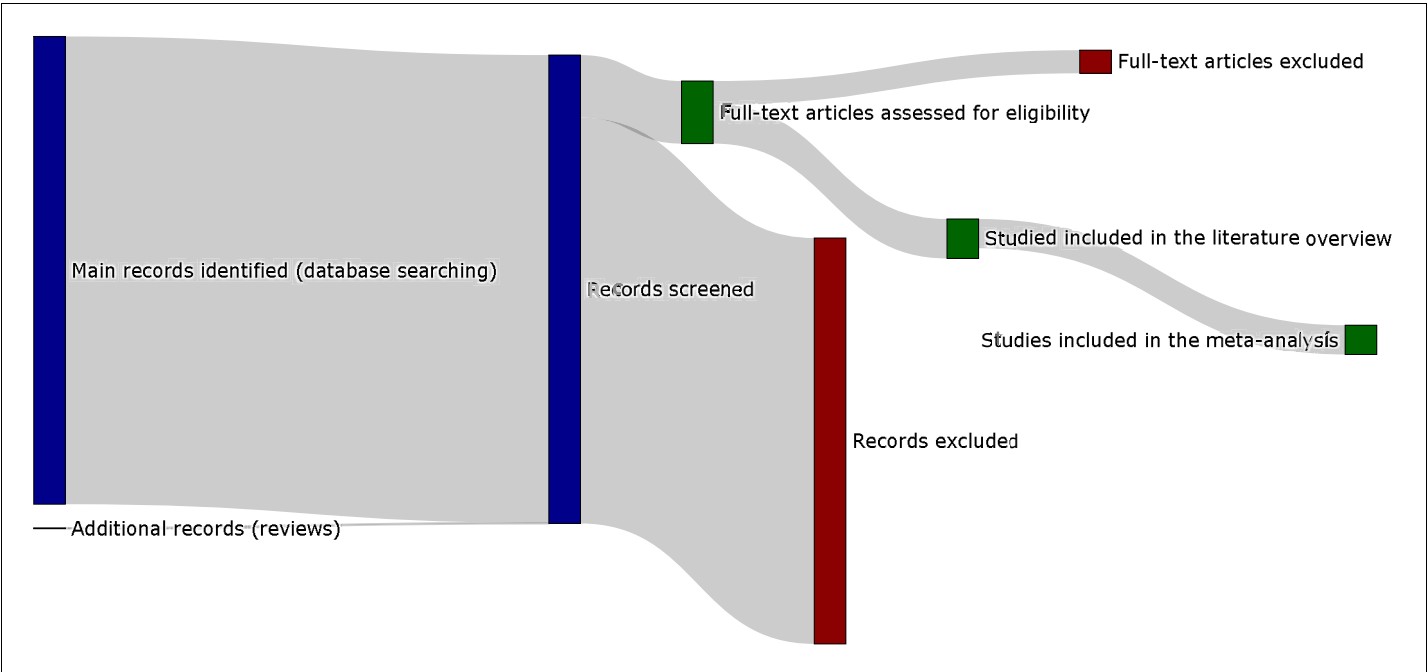

**Figure 1.** Sankey diagram representing the screening procedure (PRISMA flow chart provided in the appendix). To see the interactive figure: https://neurolibre.github.io/myelin-meta-analysis/01/selection.html#figure-1.

To combine the results for each measure, we then used a mixed-effect model: in this way we were able to express the overall effect size in terms of a range of $R^2$ values within a confidence interval, but also to assess prediction intervals and inter-study differences. The results are shown as forest plots in *Figure 4* (and also Figure S5).

Apart from MPF and MWF, all the measures showed $R^2$ overall estimates in the range 0.21–0.53. To investigate the significance of the differences between measures, we conducted a repeated measures meta-regression on every $R^2$ estimate recorded (98 in total over 43 studies). As shown in *Figure 5* (and also Figure S6), the measures can be roughly subdivided in two groups: MT- and relaxometry-based measures gave significantly higher $R^2$ estimates compared to diffusion-based measures. Within the diffusion-based measures, FA shows slightly higher estimates than the others, with marginal significance over RD and AD or no significance in case of MD.

Within MT- and relaxometry-based measures, the trends follow those in the forest plots (*Figure 4*), but most differences are not significant (*Figure 5*). However, the results in terms of z-score give a measure of distance between the $R^2$ distributions. From this perspective, MPF has higher $R^2$

**Table 1.** Results from the mixed-effect models: for each measure we reported the number of studies, the estimate and standard error of the overall $R^2$ distribution, the $\tau^2$ and the $I^2$.

| Measure | Number of studies | Estimate | Standard error | Tau² | I² |
|---------|------------------|----------|----------------|------|-----|
| MTR | 16 | 0.508 | 0.0691 | 0.07 | 96.03% |
| MPF | 10 | 0.7657 | 0.0455 | 0.0128 | 83.18% |
| FA | 17 | 0.3766 | 0.0663 | 0.0652 | 87.49% |
| RD | 15 | 0.3364 | 0.0679 | 0.0615 | 92.30% |
| MD | 12 | 0.2639 | 0.0679 | 0.044 | 87.35% |
| T1 | 8 | 0.5321 | 0.0692 | 0.0328 | 86.51% |
| AD | 9 | 0.2095 | 0.0802 | 0.048 | 97.69% |
| T2 | 7 | 0.3938 | 0.1023 | 0.0651 | 84.49% |
| MWF | 4 | 0.6997 | 0.0432 | 0.0041 | 73.19% |

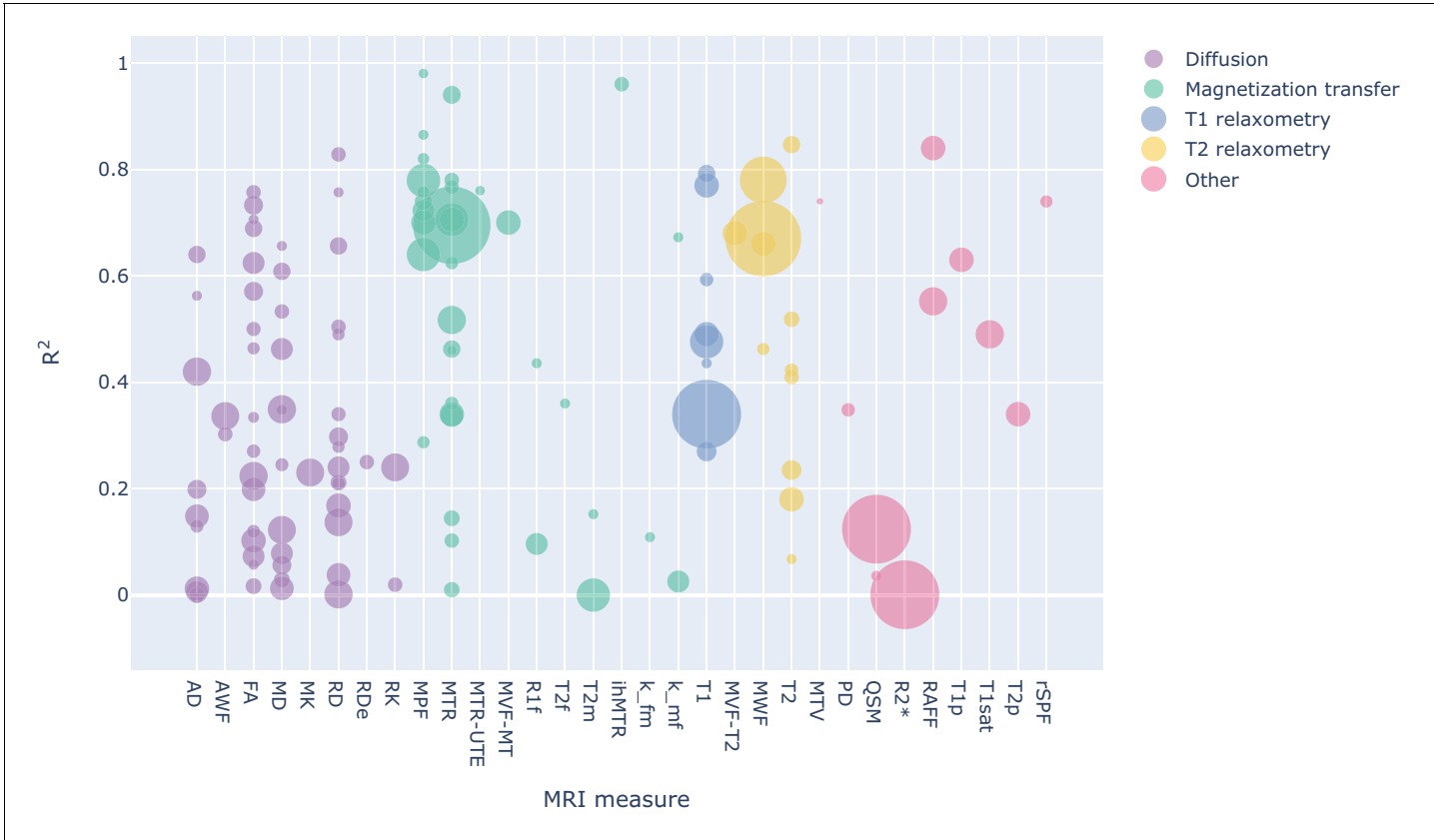

**Figure 2.** Bubble chart of $R^2$ values between a given MRI measure and histology for each study across MRI measures, with the area proportional to the number of samples. To see the interactive figure: https://neurolibre.github.io/myelin-meta-analysis/02/closer_look.html#figure-3.

estimates compared to all the other measures, but it is only marginally higher than MWF (z-score = 0.77; p-value=1) so we cannot claim that one is superior to the other. Following the same reasoning, MTR and T1 are not statistically different (z-score = 0.47; p-value=1).

When considering the prediction intervals calculated using $\tau^2$ (the variance of the effect size parameters across the population of studies), for most measures the interval spanned from 0.1 to 0.9 (*Figure 4* and Figure S5). This implies that future studies relying on such measures can expect, on the basis of these studies, to obtain any $R^2$ value in this broad interval. The only exceptions were MPF (0.49–1) and MWF (0.45–0.95), whose intervals were narrower than the alternatives. Finally, $I^2$ (a measure of how much of the variability in a typical study is due to heterogeneity in the experimental design) was generally quite high (*Table 1*). MWF showed the lowest $I^2$ across measures ($I^2$ = 73.19%), but this may be misleading considering that it was based on only four studies, while the other measures included around 10 studies. Excluding MWF, MPF also showed a relatively low $I^2$ ($I^2$ = 83.18%). Qualitative comparisons across experimental conditions and methodological choices highlighted differences across pathology models, targeted tissue types and reference techniques (*Figure 6* and Figure S7). Other factors such as magnetic field, co-registration, specific tissue and the related conditions (Figure S8) showed comparable distributions.

## Discussion

### Indirect measures are the most popular (for better or worse)

The literature survey offers an interesting perspective on popular research trends (Figure S2). The first consideration one can make is that every myelin imaging technique achieves myelin sensitivity through different means. A clear example is offered by the two most common approaches in this meta-analysis, DWI and MT: the MT effect is driven by saturation pulses interacting with myelin

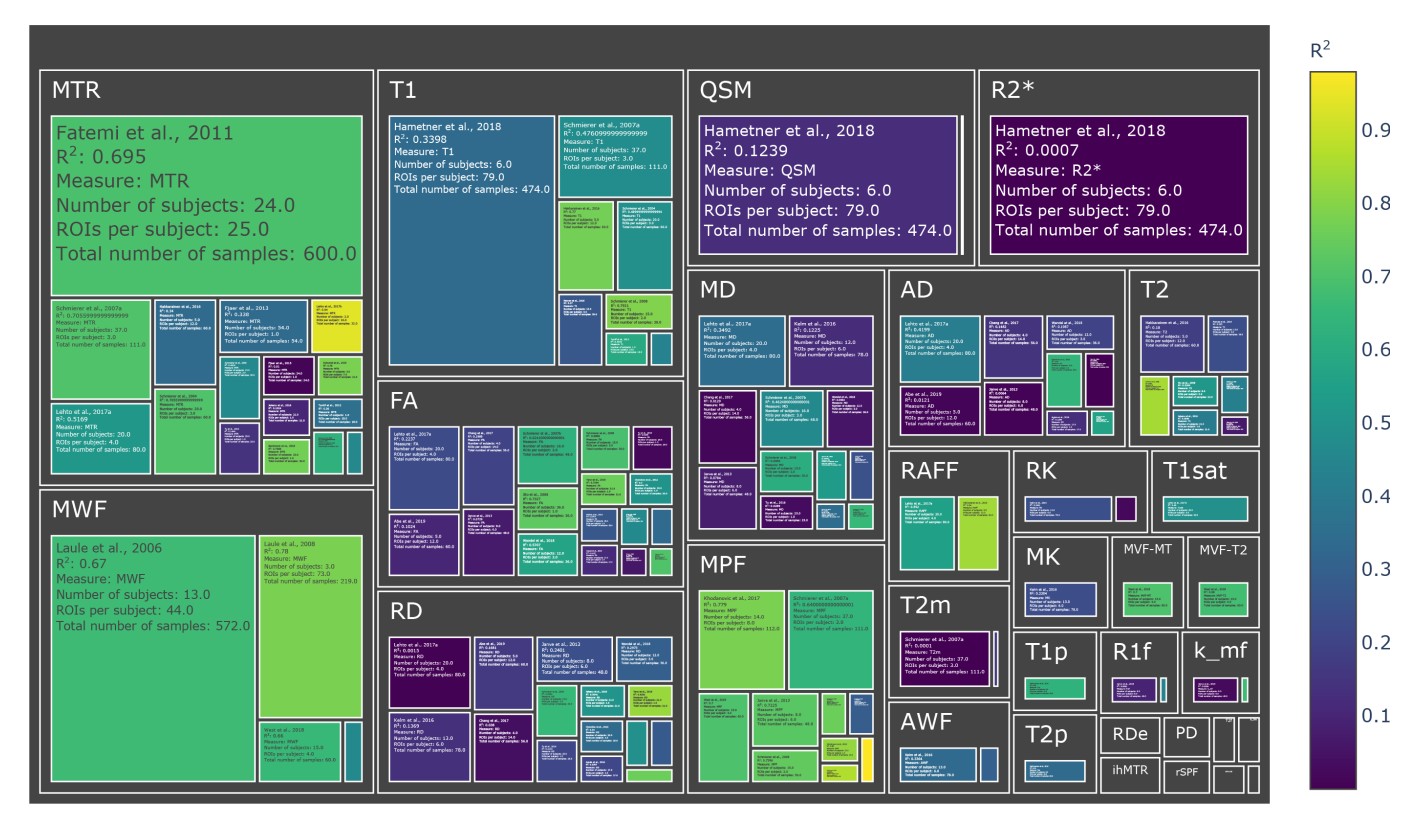

**Figure 3.** Treemap chart of the studies considered for the meta-analysis, organized by MRI measure. The color of each box represents the reported $R^2$ value while the size box is proportional to the sample size. To see the interactive figure: https://neurolibre.github.io/myelin-meta-analysis/02/closer_look.html#figure-4.

macromolecules that transfer their magnetization to water, whereas in diffusion experiments myelin is just not part of the picture. Diffusion acquisitions are blind to direct myelin measurement because the TEs used are too long (~100 ms) to be influenced by the actual macromolecules – with T2 of ~ 10 us (*Stanisz et al., 1999*) – or even the water molecules trapped in the myelin sheath – with T2 of ~ 30 ms (*MacKay et al., 1994*). To infer myelin content, one needs to rely on the interaction between intracellular and extracellular water compartments. The majority of diffusion studies included in this analysis used tensor-based measures (with fractional anisotropy being the most common), but some also used kurtosis-based analysis. The main issue with this approach is that other factors affect those measures (*Beaulieu, 2002*; *Beaulieu, 2009*), making it difficult to specifically relate changes in water compartments to changes in myelin.

Despite this issue, the use of diffusion as a proxy for myelin is quite widespread, specifically outside the field of quantitative MRI. This is probably a consequence of how popular DWI has become and how widely available are the related acquisition sequences. MT, the second most popular technique for quantifying myelin, estimates myelin by acquiring data with and without saturating the macromolecular proton pool. The simplest MT measure, MT ratio (MTR), incorporates non-myelin contributions in the final measurement. Recent acquisition variations include computing MTR from acquisitions with ultra-short echo times (*Du et al., 2009*; *Guglielmetti et al., 2020*; *Wei et al., 2018*) or relying on inhomogeneous MT (*Duhamel et al., 2019*; *Varma et al., 2015*). More complex experiments, for example quantitative MT, are based on fitting two compartments to the data, the free water and the macromolecular compartments, or pools. In this way, one is able to assess myelin through MPF with higher specificity, although still potentially including contributions from other macromolecules. Additional measures have also been considered (including the T2 of each pool, the exchange rate between the pools). The drawback of qMT is the requirement for a longer and more complex acquisition. Recently, there have been alternative techniques to estimate only MPF,

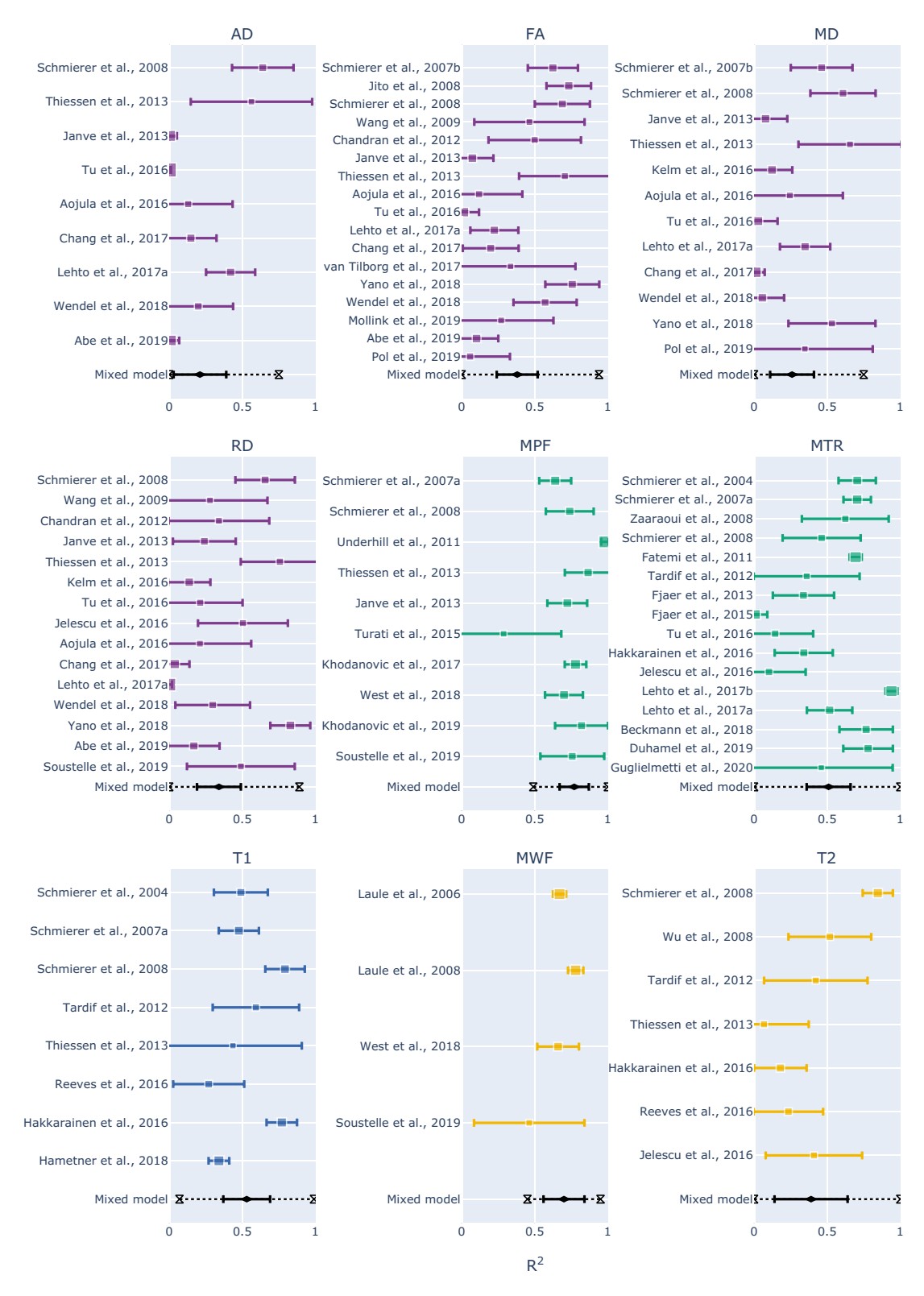

**Figure 4.** Forest plots showing the $R^2$ values reported by the studies and estimated from the mixed-effect model for each measure. The hourglasses and the dotted lines in the mixed-effect model outcomes represent the prediction intervals. To see the interactive figure: https://neurolibre.github.io/myelin-meta-analysis/03/meta_analysis.html#figure-5.

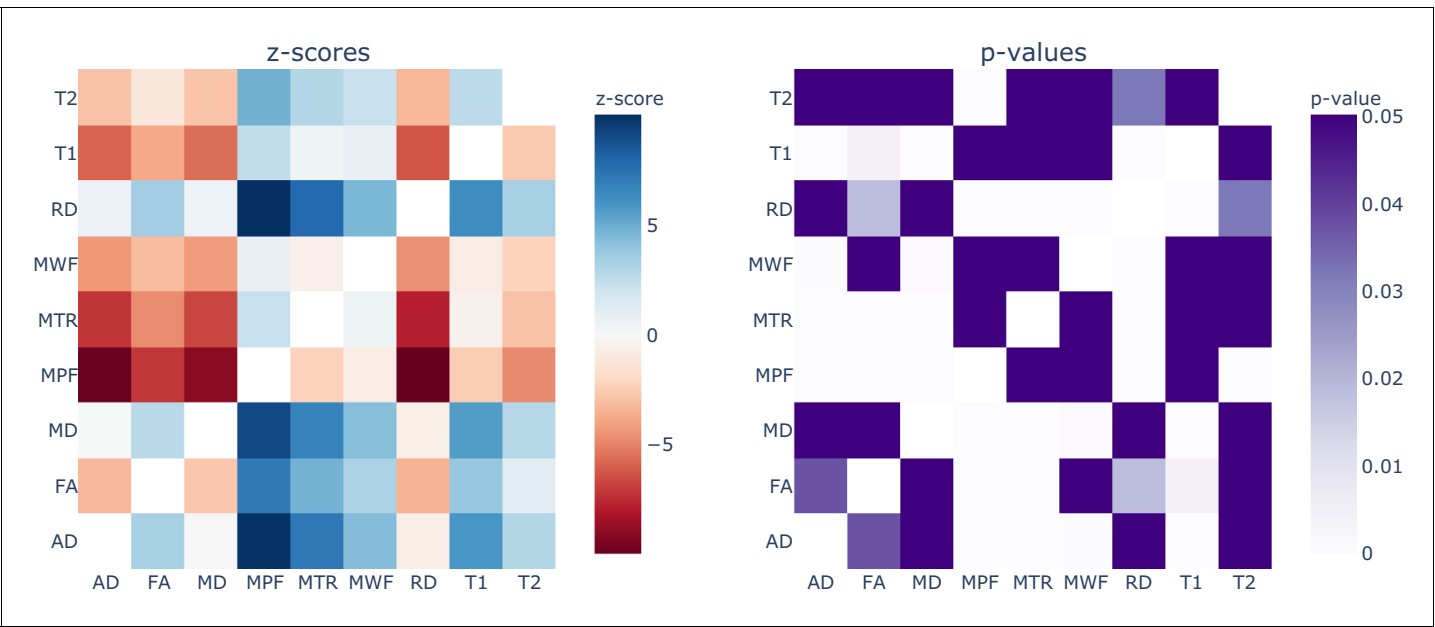

**Figure 5.** Results from the repeated measures meta-regression, displayed in terms of z-scores (left) and p-values (right) for each pairwise comparison across all the MRI measures. In the z-score heatmap, each element refers to the comparison between the measure on the x axis with the one on the y axis. For example, MPF and FA (z-score = 7.14; p-value<0.0001) are statistically different, while MPF and T1 (z-score = 2.51; p-value=0.43) are not statistically different. To see the interactive figure: https://neurolibre.github.io/myelin-meta-analysis/03/meta_analysis.html#figure-6.

resulting in faster acquisitions with similar results (*Khodanovich et al., 2019*; *Khodanovich et al., 2017*; *Yarnykh, 2012*). Despite being focused on macromolecular contributions, these approaches are not strictly specific to myelin (*Sled, 2018*): in this sense, an important limitation is that MT effects are sensitive to the pH of the targeted tissue and therefore changes in the pH (caused for example by inflammation processes) will affect MT-based measures of myelin (*Stanisz et al., 2004*).

Following diffusion and MT, the most popular approach is T2 relaxometry. Unlike diffusion and MT, in T2 relaxometry experiments one can directly observe the contribution from the water trapped between the myelin bilayers, and can therefore estimate the myelin water fraction. A simpler but less specific approach consists in estimating the transverse relaxation time considering the decay to be mono-exponential. A historical and practical drawback of these approaches is that they require longer acquisitions, although faster alternatives have been developed (*Does and Gore, 2000*; *Prasloski et al., 2012*). A more subtle but nevertheless important limitation lies in the multi-compartment model used in multi-exponential T2 relaxometry (*Does, 2018*): this model generally assumes slow water exchange between compartments, but it has been showed that water exchange actually contributes to T2 spectra variations (*Dula et al., 2010*; *Harkins et al., 2012*).

Finally, other studies used a diverse collection of other measures, including T1 relaxometry, apparent transversal relaxation rate (R2*), proton density (PD), macromolecular tissue volume (MTV), relaxation along a fictitious field (RAFF), and quantitative susceptibility mapping (QSM).

After this general overview, it is clear that each modality could be a suitable candidate for a quantitative myelin biomarker. To then make a choice informed by the studies here reported, it becomes necessary to consider not only effect sizes in terms of correlation, but also sample sizes and acquisition times.

## There is no myelin MRI measure true to histology

When looking at the $R^2$ values across the different measures, the first detail that catches one's eye is how most measures present a broad range of values (*Figure 2* and *Figure 3*). When taking into account the sample size, the largest studies show higher correlations for MT and T2 relaxometry studies than any other approach (Figure S3 and Figure S4). In quantitative terms, the meta-analysis corroborates this idea, showing that MPF and MWF tend to be more specific to myelin compared to the other measures (respectively with $R^2 = 0.7657$ and $R^2 = 0.6997$), in line with the underlying

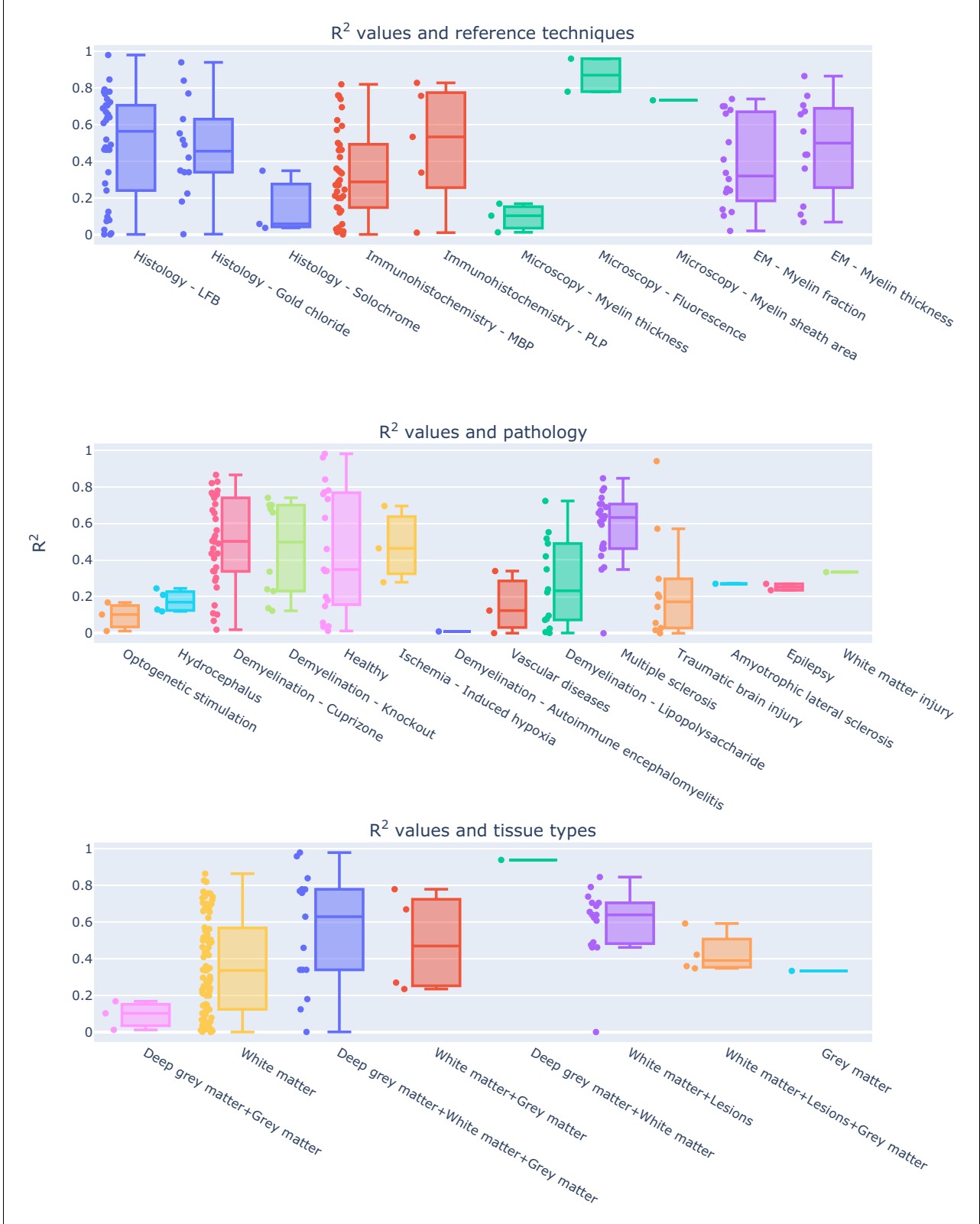

**Figure 6.** Experimental conditions and methodological choices influencing the $R^2$ values (top: reference techniques; middle: pathology model; bottom: tissue types). To see the interactive figure: https://neurolibre.github.io/myelin-meta-analysis/04/other_factors.html#figure-7.

theory. Notably, diffusion-based measures show the lowest overall estimates (with values between $R^2 = 0.3766$ for FA and $R^2 = 0.2095$ for AD): this could be due to the fact, as already mentioned, that DWI does not specifically measure myelin properties, and despite FA and RD being influenced by the myelin content, they are also influenced by other factors that make them unsuitable as measures of myelin. The repeated measure meta-regression confirms this overall picture, clearly distinguishing MT- and relaxometry-based measures from diffusion-based ones (*Figure 5*).

Despite these considerations on the advantages of MPF and MWF, one should refrain from concluding that they are the 'true' MRI measures of myelin. The reason for this caution is given not by the overall effect sizes observed here, but by the collateral outcomes of the meta-analysis. The first one is given by the prediction intervals: most measures exhibit large intervals (*Figure 4*), not supporting the idea of them being robust biomarkers. MPF and MWF seem to be again the most suitable choices for future studies, but a range between 0.5 and 1 is still quite large.

The second important aspect to consider is given by the differences across studies: the meta-analysis showed how such differences strongly limit inter-study comparisons for a given measure (*Figure 6*). This result should be expected, given that the studies here examined are inevitably influenced by the specific experimental constraints and methodological choices. Given the limited number of studies, it is not possible to quantitatively study interactions between MRI measures and the other factors (e.g. modality used as a reference, tissue types, magnetic field strength). For further qualitative insights, we invite the reader to explore the interactive figures S7-S8. A first important factor to consider is the validation modality used as a reference, which will be dictated by the equipment availability and cost. However, such a choice has an impact on the actual comparison: histology and immunochemistry, despite being specific to myelin, do not offer a volumetric measure of myelin, but rather a proxy based on the transmittance of the histological sections. So far, the only modality able to give a volumetric measure would be electron microscopy, which is an expensive and resource-consuming approach. Also, electron microscopy has several limitations, including tissue shrinkage, degradation of the myelin sheath structure due to imperfect fixation, imperfect penetration of the osmium stain, polishing, keeping focus over large imaging regions. All these effects contribute to the lack of precision and accuracy when quantifying myelin content with EM-based histology (*Cohen-Adad, 2018*). Another important observation is that none of the studies here reviewed considered histology reproducibility, which is hard to quantify as a whole given that a sample can be processed only once: collateral factors affecting tissue processing (e.g. sectioning distortions, mounting and staining issues) constitute an actual limitation for histology-based validation. A further example of influential factor often dictated by equipment availability is the magnetic field strength of the MRI scanner: figure S8 shows that most studies were conducted at 7T and 9.4T, with some pioneering studies at 1.5T and even fewer ones at other field strengths.

In addition to differences in experimental and methodological designs, there are also several considerations that arise out of the lack of shared practices in MRI validation studies. The first evident one is the use of correlations: despite being a simple measure that serves well the purpose of roughly characterizing a relationship, Pearson correlation is not the right tool for quantitative biomarkers, as it does not characterize the actual relationship between histology and MRI. Linear regression is a step forward but has the disadvantage of assuming a linear relationship. Despite Pearson correlation and linear regression being the most common measures used in the studies here reviewed, it is still not clear if the relationship is actually linear. Only one study among the considered ones computed both Pearson and Spearman correlation values (*Tardif et al., 2012*), and reported higher Spearman correlations, pointing out that non-linear relationships should actually be considered. One last consideration regarding the use of correlation measures for validating quantitative biomarkers is about the intercept in the MRI-histology relationship. Notably, only MWF is expected to assume a value equal to zero when myelin is absent (*West et al., 2018*). For the other measures, it would be necessary to estimate the intercept, which leads to the calibration problem in the estimate of myelin volume fraction. Notably, calculating Pearson correlation does not provide any information for such calibration. Another arbitrary practice that would benefit from some harmonization is the choice of ROIs. The studies reported here examined a diverse list of ROIs, in most cases hand-drawn on each modality, encompassing different types of tissue, and the most common approach is to report a single, pooled correlation. This is problematic, as different types of tissue (e.g. grey matter and white matter) will show different values for MRI-based measures but also for histology-based ones, making linearity assumptions about the two modalities. However, with this

approach gross differences between tissues drive the observed correlation, without actually showing if the MRI-based measure under analysis is sensitive to subtle differences and therefore a suitable quantitative biomarker for myelin. The effect of considering different types of tissues is showed in *Figure 6* and Figure S7, where correlation ranges change when considering different types of tissue. However, the large correlation range in white matter, the most common tissue studied, suggests that other factors also affect the correlation.

It should be clear at this point that any debate about a universal MRI-based measure of myelin is pointless, at least at the moment, as the overall picture provided by previous studies does not point to any such ideal measure. Nevertheless, is debating about a universal measure helpful for future studies?

## Better biomarkers require more reproducibility studies

We hope this meta-analysis convinces the reader that a holy grail of myelin imaging does not exist, at least as long as we consider histology to be the ground truth. Given that we all have to pick our poison, the upside is that measures based on MT and relaxometry are not statistically different, and therefore, future studies have an actual choice among candidate measures. For further progress, rather than debating about a perfect measure, we would argue that what is missing at the moment is a clear picture of what can be achieved with each specific MRI modality. The studies examined here focus on a large set of different measures, and more than half of them considered at most two measures, highlighting how the field is mostly focused on formulating new measures. While it is understood that novel measures can provide new perspectives, it is also fundamentally important to understand the concrete capabilities and limitations of current measures. From this meta-analysis, what the literature clearly lacks is reproducibility studies, specifically answering two main questions: (1) what is the specificity of each measure? We should have a practical validation of our theoretical understanding of the relevant confounds; (2) what is the 'parameter sensitivity' of each measure? Here, we refer to parameter sensitivity in a broad sense, that includes also experimental conditions and methodological choices. The results here presented show how certain conditions (e.g. pathology) seem to affect the coefficient of determination more than others but given the limited number of studies for each modality, we refrained from additional analyses to avoid speculation. A warning message that is evident from these results is the inherent limitation of DWI for estimating myelin content: this is not by any means a novel result (*Beaulieu, 2002*; *Beaulieu, 2009*), but it is nevertheless worth reiterating given the outcomes of our analysis. If estimating myelin content is relevant in a diffusion study, it is important to consider complementing the diffusion measure with one of the modalities here reviewed; in this way, it would be possible to decouple the influence of myelin content from the many other factors that come into play when considering diffusion phenomena.

Finally, an important factor to take into account when choosing a biomarker of myelin is the actual application. For animal research, long acquisitions are not a major issue. However, when considering biomarkers for potential clinical use, the acquisition time can become a relevant issue. An example is the well-established multi-echo spin-echo implementation of MWF, that can only be used for a specific slice in a hypothetical clinical scenario. Faster techniques have been proposed for estimating it with gradient- and spin-echo (GRASE) sequences (*Does and Gore, 2000*; *Feinberg and Oshio, 1991*; *Prasloski et al., 2012*). Even in this case, the acquisition time still reaches 15 min for acquiring roughly the whole brain with an isotropic resolution of 2 mm. Complex MT acquisitions such as qMT suffer from the same problem, although it is possible to use optimized and faster protocols to focus specifically on MPF (*Khodanovich et al., 2019*; *Khodanovich et al., 2017*; *Yarnykh, 2012*).

## Conclusions

Several MRI measures are sensitive to myelin content and the current literature suggests that most of them are not statistically different in terms of their relationship with the underlying histology. Measures highly correlated with histology are also the ones with a higher expected specificity. This suggests that future studies should try to better address how specific each measure is, for the sake of clarifying suitable applications.

## Materials and methods

### Review methodology

The Medline database (https://pubmed.ncbi.nlm.nih.gov) was used to retrieve the articles. The keywords used are specified in the appendix. We followed the PRISMA (Preferred Reporting Items for Systematic Reviews and Meta-Analyses) guidelines for record screening and study selection. The results were first screened to remove unrelated work. Specifically we discarded: work relying only on MRI; work relying only on histology or equivalent approaches; work reporting only qualitative comparisons. After this first screening, the remaining papers were assessed. At this stage, we discarded: studies using MRI-based measures in arbitrary units (e.g. T1-weighted or T2-weighted data); studies using measures of variation in myelin content (defined either as the difference between normal and abnormal myelin content) either for MRI or for histology; studies using arbitrary assessment scales; studies comparing MRI-based absolute measures of myelin with histology-based relative measures (e.g. g-ratio); studies reporting other quantitative measures than correlation or $R^2$ values; studies comparing histology from one dataset and MRI from a different one. As an additional source for potential candidate studies, we screened the review articles in the initial results, and we selected the relevant studies that were not already present in the studies already selected.

From the final papers, we collected first the following details: the DOI; which approach was used (diffusion, MT, T1 relaxometry, T2 relaxometry, or other); which specific MRI measures were compared to histology or equivalent techniques; the magnetic field; the technique used as a reference (histology, immunochemistry, microscopy, electron microscopy); the focus of the study in terms of brain, spinal cord or peripheral nerve; if the subjects were humans or animals, and if the latter which animal; if the tissue under exam was in vivo, in situ or ex vivo, and in the latter case if the tissue was fixed or not; if the tissue was healthy or pathological, and if the latter which pathology; the specific structures examined for correlation purposes; which comparison technique was used (e.g. Pearson correlation, Spearman correlation, linear regression); the number of subjects; the number of ROIs per subject; the male/female ratio; if registration procedures were performed to align MRI and histology; in case of pathological tissue, if control tissue was considered as well; other relevant notes. If before calculating the correlations the data were averaged across subjects, the number of subjects was considered to be one. The same consideration was made for averaging across ROIs. This is because the numbers of subjects and ROIs were used to take into account how many sample points were used when computing the correlation. We set each of those numbers to one for all the studies where the data were averaged respectively across subjects and across ROIs. Finally, in those cases where the number of ROIs or the number of subjects were given as a range rather than specific values, we used the most conservative value and added the related details to the notes.

We then proceeded to collect the quantitative results reported for each measure and for each study in the form of $R^2$. Given that different studies may rely on a different strategy when reporting correlations, we adopted the following reasoning to limit discrepancies across studies while still objectively representing each of them. In case of multiple correlation values reported, for our analysis we selected the ones referring to the whole dataset and the entire brain if available, and considering each ROI in a given subject as a sample if possible; if only correlation values for specific ROIs were reported, the one for the most common reported structure would be chosen. In the case of multiple subjects, if data were provided separately for each group, the correlation for the control group was used. When different comparison methods were reported (e.g. both Pearson and Spearman correlation) or if the MRI data was compared with multiple references (e.g. both histology and immunohistochemistry), the correlations used were chosen on the basis of the following priority orders (from the most preferable to the least): for multiple comparison methods, linear regression, Spearman correlation, Pearson correlation; for multiple reference techniques, electron microscopy, immunohistochemistry, histology. Finally, in any other case where more than one correlation value was available, the most conservative value was used. Any other additional value was in any case mentioned in the notes of the respective study.

### Meta-analysis

For the quantitative analysis, we restricted our focus on brain studies and only on the ones providing an indication of both the number of subjects and the number of ROIs. For each study, we computed

the sample size as the product between the number of subjects and the number of ROIs per subject. In this way, we were able to compare the reported $R^2$ values across measures taking into account the related number of points actually used for correlation purposes. We note that correlation or regression analyses run on multiple ROIs and subjects represents a repeated measures analysis, for which the degrees of freedom computation can be complex; however, most papers neglected the repeated measures structure of the data and thus the sample size computation here represents a very approximate and optimistic view of the precision of each $R^2$ value.

To estimate the variance of each $R^2$ value, we relied on the correlation properties and the delta method (*Lehman, 1999*). Let us consider the Pearson's correlation r of two variables X and Y with population correlation ρ. If r is calculated from N random samples, the sampling variance is $(1-\rho^2)^2/$N. Applying the delta method, we then approximated the variance of $R^2$ as $4 R^2(1 R^2)^2/N$, assuming $R^2 \approx \rho^2$. As we recognise that some papers computed Spearman correlation, this calculation is again optimistic and may underestimate the sampling variability of the squared Spearman correlation.

To estimate the overall effect size in terms of $R^2$, we have to choose how to model the distribution of true effects given by the data collected from the literature. The two most common approaches are fixed-effects and mixed-effects models. While the underlying mathematical model is the same as the one used for linear regression (more details in the appendix), the assumptions are different: fixed-effects models assume that all the studies share a common effect size, while mixed-effects models assume that the effect size across studies is similar but not identical (*Raudenbush, 2009*). In our case, as the studies have several factors that influence the $R^2$ values (e.g. histology/microscopy reference, magnetic field strength, pathology model), we expect a distribution of effect sizes due to inter-study differences. This is why we proceeded to fit a mixed-effects model to each measure that was featured in more than two studies. Apart from the effect size distributions, we reported two additional measures, $I^2$ and $\tau^2$: the former expresses as a percentage how much of variability in a typical study is due to heterogeneity (i.e. the variation in study outcomes between studies) rather than chance (*Higgins and Thompson, 2002*), while the latter can be used to calculate the prediction interval (*Raudenbush, 2009*), which gives the expected range for the measure of interest in future studies. We used forest plots to represent the outcomes, and both the mixed effects estimate of the population estimated $R^2$, with both a 95% confidence and a (larger) 95% prediction interval.

For the explicit purpose of comparing the effect sizes between different MRI measures, we conducted a repeated measures meta-regression on every $R^2$ value recorded. We associated each $R^2$ value with three additional details: (i) the related variance, as done in the measure-specific mixed-effects models; (ii) the related study, used as the random intercept (i.e. random variable) to incorporate potential inter-study variability; and (iii) the related MRI measure, used as the moderator (i.e. categorical variable) to estimate the differences between measures. In this way, the meta-regression leads to $R^2$ intervals for each MRI measure, with the same trend as measure-specific mixed-effects models but with subtle differences. This is because the meta-regression makes two additional assumptions: first, $R^2$ estimates within the same study share the same random effects and second, the between-study variance is the same for all observations. We then used the meta-regression $R^2$ estimates to compute every possible pairwise comparison between MRI measures and to identify significantly different pairs using Tukey's test, while controlling the error rate over all the possible comparisons (Bonferroni correction).

This additional model is necessary, as direct comparisons are not possible with measure-specific analyses. While the repeated measures meta-regression makes direct comparisons straightforward, we reported the main $R^2$ estimates based on the measure-specific mixed-effects models, as they make weaker assumptions.

For visual comparisons, we used the Jupyter notebook provided in the supplementary materials. For model fitting, we used the Metafor package, version 2.4–0 (*Viechtbauer, 2010*).

## Acknowledgements

MM was funded by the Wellcome Trust through a Sir Henry Wellcome Postdoctoral Fellowship [213722/Z/18/Z]. TEN was supported by NIH grant R01MH096906.

## Additional information

### Funding

| Funder | Grant reference number | Author |
| --- | --- | --- |
| Wellcome Trust | 213722/Z/18/Z | Matteo Mancini |
| National Institutes of Health | R01MH096906 | Thomas E Nichols |

The funders had no role in study design, data collection and interpretation, or the decision to submit the work for publication.

### Author contributions

Matteo Mancini, Conceptualization, Data curation, Formal analysis, Visualization, Methodology, Writing - original draft, Writing - review and editing; Agah Karakuzu, Visualization, Writing - review and editing; Julien Cohen-Adad, Mara Cercignani, Conceptualization, Writing - review and editing; Thomas E Nichols, Nikola Stikov, Conceptualization, Methodology, Writing - review and editing

### Author ORCIDs

Matteo Mancini ⓘ https://orcid.org/0000-0001-7194-4568
Agah Karakuzu ⓘ https://orcid.org/0000-0001-7283-271X
Julien Cohen-Adad ⓘ https://orcid.org/0000-0003-3662-9532
Mara Cercignani ⓘ https://orcid.org/0000-0002-4550-2456
Thomas E Nichols ⓘ https://orcid.org/0000-0002-4516-5103
Nikola Stikov ⓘ http://orcid.org/0000-0002-8480-5230

### Decision letter and Author response

Decision letter https://doi.org/10.7554/eLife.61523.sa1
Author response https://doi.org/10.7554/eLife.61523.sa2

## Additional files

### Supplementary files

• Source code 1. A Jupyter notebook in ipynb format containing the Python code used to process the data, run the analyses and generate all the figures. In order to execute the notebook, the Python (3.7) and R (3.6) interpreters are required, as well as the R packages metafor (2.4) and multcomp (1.4), and the following Python packages: numpy (1.18.4); pandas (0.25.3); plotly (4.8.1); rpy2 (3.3.4); xlrd (1.2.0). The notebook assumes that the spreadsheet is in the same path as the notebook itself. More details are provided here: https://github.com/matteomancini/myelin-meta-analysis (*Mancini, 2020*; copy archived at swh:1:rev: 17ca8673c9e15c54ad0b814248b69232b63c3a38).

• Source data 1. A spreadsheet in xlsx format containing all the data and details collected for the studies considered in this systematic review.

• Supplementary file 1. A multimedia file in HTML format containing an interactive version of the figures in this manuscript plus additional ones.

• Transparent reporting form

### Data availability

All the data collected from the selected studies for this meta-analysis are provided in the spreadsheet file Source data 1.

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

## Appendix 1

### Search keywords

(myelin[Title/Abstract] AND ((magnetic[Title/Abstract] AND resonance[Title/Abstract]) OR mr[Title/Abstract] OR mri[Title/Abstract])) AND (histology[Title/Abstract] OR histopathology[Title/Abstract] OR microscopy[Title/Abstract] OR immunohistochemistry[Title/Abstract] OR histological[Title/Abstract] OR histologically[Title/Abstract] OR histologic[Title/Abstract] OR histopathological[Title/Abstract] OR histopathologically[Title/Abstract] OR histopathologic[Title/Abstract]).

Results obtained from the Medline database: 688 (03/06/2020).

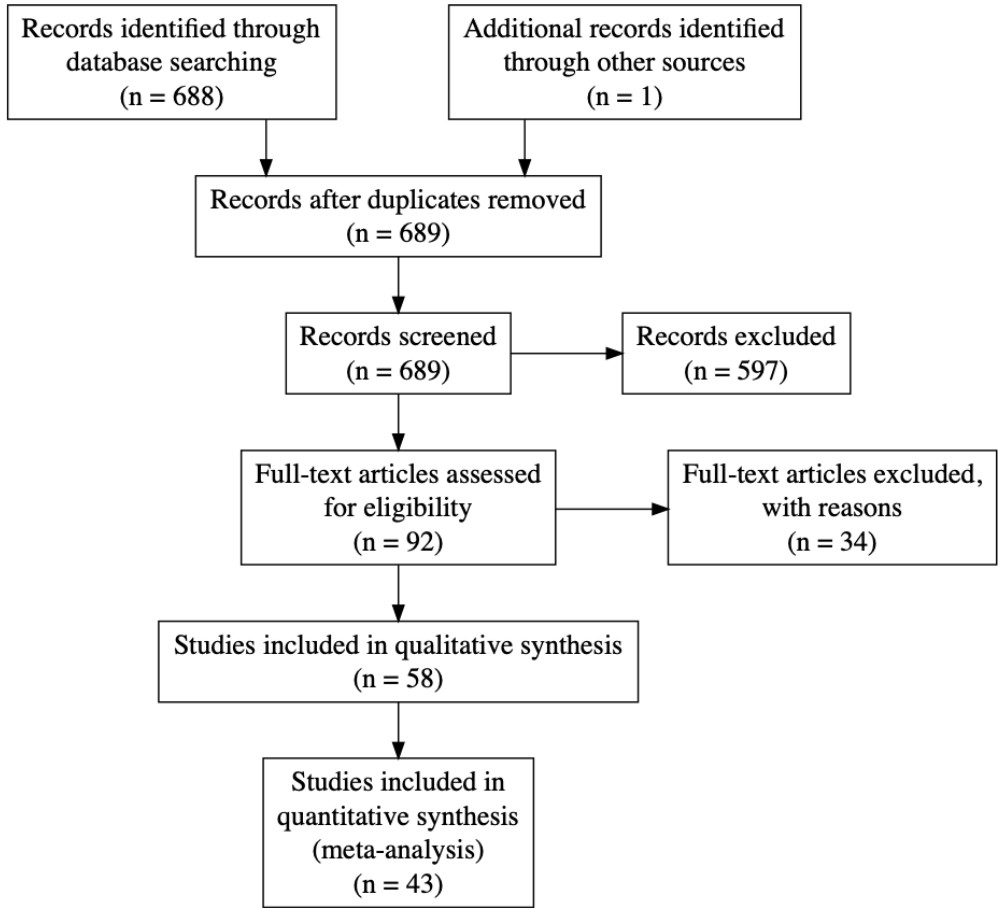

**Appendix 1—figure 1.** PRISMA flowchart for the meta-analysis.

### Fixed- and mixed-effects models

While a traditional linear regression model estimates the error variance from residuals, in a fixed effects meta-analysis model, each paper's response and standard errors, as well as the error variance of the regression model can be directly computed from the supplied response standard deviations. Specifically, for a (non-meta) regression model we have the $i$-th response $y_i$ modeled with covariate values $x_i$, $y_i = x_i\beta + \varepsilon_i$, where random error has unknown variance $\mathrm{Var}(\varepsilon_i) = \sigma^2$. In a fixed-effects meta-analysis, we are given $y_i$ but also $s_i$, the standard error of $y_i$, and the regression model has the same form except the variance is known, $\mathrm{Var}(\varepsilon_i) = s_i^2$, and the weighted least squares regression can be computed, estimating beta and its standard error. A mixed-effects meta-analysis accounts for more variance than what can be ascribed to the sampling error of the reported outcome. The regression model has again the same form, except now the variance is $\mathrm{Var}(\varepsilon_i) = s_i^2 + \tau^2$, the sum of the reported squared standard error and the unknown between-study variance $\tau^2$. Iterative methods are used to estimate $\tau^2$ and, once estimated, a weighted least squares regression can be computed. The parameter $\tau^2$ can be interpreted as the variance of *noise-free* (hypothetical, zero standard error)

results from the population of all possible studies. The importance of $\tau^2$ can also be gauged by $I^2$, the proportion of variance due to random inter-study differences (i.e. $1 - I^2$ is the proportion attributable to random sampling error of each study) (*Higgins and Thompson, 2002*).

## Abbreviations and mathematical symbols

AD – axial diffusivity
AK – axial kurtosis
AWF – axonal water fraction
FA – fraction anisotropy
ihMTR – inhomogeneous magnetization transfer ratio
k_fm – free water-macromolecular exchange rate
k_mf – macromolecular-free water exchange rate
M0m – macromolecular pool magnetization fraction
MD – mean diffusivity
MK – mean kurtosis
MPF – macromolecular pool fraction
MT – magnetization transfer
MTR – magnetization transfer ratio
MTR-UTE – magnetization transfer ratio (using ultra-short echo time)
MTV – macromolecular tissue volume
MVF-MT – myelin volume fraction (estimated from MT)
MVF-T2 – myelin volume fraction (estimated from T2)
MWF – myelin water fraction
PD – proton density
PN – peripheral nerve
PRISMA – Preferred Reporting Items for Systematic Reviews and Meta-Analyses
QSM – quantitative susceptibility mapping
R1f – free water pool longitudinal relaxation rate
R2* – apparent transverse relaxation rate
RAFF – relaxation along a fictitious field
RD – radial diffusivity
RD-DBSI – radial diffusivity (from diffusion basis spectrum imaging)
RDe – extra-cellular compartment radial diffusivity
RK – radial kurtosis
rSPF – relative semi-solid proton fraction
SC – spinal cord
T1 – longitudinal relaxation time
T1p – adiabatic longitudinal relaxation time
T1sat – longitudinal relaxation time under magnetization transfer irradiation
T2 – transverse relaxation time
T2f – free water pool transverse relaxation time
T2int – transverse relaxation intermediate component
T2m – macromolecular pool transverse relaxation rate
T2p – adiabatic transverse relaxation time

**Appendix 1—table 1.** Selected studies for qualitative analysis.

| Study | MRI measure(s) | Histology/microscopy measure | Tissue | Condition | Focus |
|---|---|---|---|---|---|
| *Schmierer et al., 2004* | T1, MTR | Histology - LFB | Human | Multiple sclerosis | Brain |
| *Odrobina et al., 2005* | T1, T2, T2int, MWF, M0m, MTR | Microscopy - Myelin fraction | Animal - Rat | Demyelination - Tellurium | PN |
| *Pun et al., 2005* | T1, T2int, MWF | Microscopy - Myelin fraction | Animal - Rat | Demyelination - Tellurium | PN |

*Continued on next page*

*Appendix 1—table 1 continued*

| Study | MRI measure(s) | Histology/microscopy measure | Tissue | Condition | Focus |
|---|---|---|---|---|---|
| *Laule et al., 2006* | MWF | Histology - LFB | Human | Multiple sclerosis | Brain |
| *Schmierer et al., 2007a* | T1, MTR, MPF, T2m | Histology - LFB | Human | Multiple sclerosis | Brain |
| *Schmierer et al., 2007b* | FA, MD | Histology - LFB | Human | Multiple sclerosis | Brain |
| *Jito et al., 2008* | FA | Microscopy - Myelin sheath area | Animal - Mouse | Healthy | Brain |
| *Kozlowski et al., 2008* | MWF, FA, AD, RD, MD | Immunohistochemistry - MBP | Animal - Rat | Injury - Dorsal columnar transection | SC |
| *Laule et al., 2008* | MWF | Histology - LFB | Human | Multiple sclerosis | Brain |
| *Schmierer et al., 2008* | T1, T2, MTR, MPF, MD, FA, AD, RD | Histology - LFB | Human | Multiple sclerosis | Brain |
| *Wu et al., 2008* | T2 | Histology - LFB | Animal - Mouse | Demyelination - Cuprizone | Brain |
| *Zaaraoui et al., 2008* | MTR | Immunohistochemistry - MBP | Animal - Mouse | Demyelination - Cuprizone | Brain |
| *Takagi et al., 2009* | FA, AD | EM - Myelin thickness | Animal - Rat | Degeneration - Contusive injury | PN |
| *Wang et al., 2009* | FA, RD | Histology - LFB | Animal - Rat | Ischemia - Induced hypoxia | Brain |
| *Zhang et al., 2009* | RD | Histology - LFB | Animal - Rat | Injury - Dorsal columnar transection | SC |
| *Schmierer et al., 2010* | MTR, T2 | Histology - LFB | Human | Multiple sclerosis | Brain |
| *Fatemi et al., 2011* | MTR | Immunohistochemistry - MBP | Animal - Mouse | Ischemia - Induced hypoxia | Brain |
| *Laule et al., 2011* | MWF | Immunohistochemistry - MBP | Human | Multiple sclerosis | Brain |
| *Underhill et al., 2011* | MPF | Histology - LFB | Animal - Mouse | Healthy | Brain |
| *Chandran et al., 2012* | FA, RD | Immunohistochemistry - MBP | Animal - Mouse | Demyelination - Cuprizone | Brain |
| *Tardif et al., 2012* | T1, T2, MTR, PD | Immunohistochemistry - MBP | Human | Multiple sclerosis | Brain |
| *Fjær et al., 2013* | MTR | Immunohistochemistry - PLP | Animal - Mouse | Demyelination - Cuprizone | Brain |
| *Harkins et al., 2013* | MWF, MPF | Microscopy - Myelin fraction | Animal - Rat | Edema - Hexaclorophene | SC |
| *Janve et al., 2013* | MPF, R1a, k_ba, FA, RD, MD, AD | Histology - LFB | Animal - Rat | Demyelination - Lipopolysaccharide | Brain |
| *Thiessen et al., 2013* | MPF, R1f, k_fm, k_mf, T2f, T2m, MD, RD, AD, FA, T1, T2 | EM - Myelin thickness | Animal - Mouse | Demyelination - Cuprizone | Brain |
| *Kozlowski et al., 2014* | MWF | Immunohistochemistry - MBP | Animal - Rat | Injury - Dorsal columnar transection | SC |
| *Wang et al., 2014* | RD, RD-DBSI | Immunohistochemistry - MBP | Animal - Mouse | Demyelination - Autoimmune encephalomyelitis | SC |

*Appendix 1—table 1 continued*

| Study | MRI measure(s) | Histology/microscopy measure | Tissue | Condition | Focus |
|---|---|---|---|---|---|
| *Fjær et al., 2015* | MTR | Immunohistochemistry - PLP | Animal - Mouse | Demyelination - Autoimmune encephalomyelitis | Brain |
| *Seehaus et al., 2015* | FA, RD, MD | Histology - Silver | Human | Healthy | Brain |
| *Turati et al., 2015* | MPF | Immunohistochemistry - MBP | Animal - Mouse | Demyelination - Cuprizone | Brain |
| *Wang et al., 2015* | RD-DBSI | Histology - LFB | Human | Multiple sclerosis | SC |
| *Aojula et al., 2016* | FA, AD, RD, MD | Immunohistochemistry - MBP | Animal - Rat | Hydrocephalus | Brain |
| *Hakkarainen et al., 2016* | T1, T2, MTR, T1p, T2p, RAFF | Histology - Gold chloride | Animal - Rat | Healthy | Brain |
| *Jelescu et al., 2016* | RD, RK, AWF, Rde, T2, MTR | EM - Myelin fraction | Animal - Mouse | Demyelination - Cuprizone | Brain |
| *Kelm et al., 2016* | MD, RD, MK, RK, AWF | EM - Myelin fraction | Animal - Mouse | Demyelination - Knockout | Brain |
| *Reeves et al., 2016* | T1, T2 | Immunohistochemistry - MBP | Human | Epilepsy | Brain |
| *Tu et al., 2016* | FA, AD, RD, MD, MTR | Immunohistochemistry - MBP | Animal - Rat | Traumatic brain injury | Brain |
| *Chang et al., 2017* | FA, AD, RD, MD | Immunohistochemistry - MBP | Animal - Mouse | Healthy | Brain |
| *Chen et al., 2017* | MWF | EM - Myelin fraction | Animal - Rat | Injury - Dorsal columnar transection | SC |
| *Khodanovich et al., 2017* | MPF | Histology - LFB | Animal - Mouse | Demyelination - Cuprizone | Brain |
| *Lehto et al., 2017a* | RAFF, MTR, T1sat, FA, MD, AD, RD | Histology - Gold chloride | Animal - Rat | Demyelination - Lipopolysaccharide | Brain |
| *Lehto et al., 2017b* | MTR | Histology - Gold chloride | Animal - Rat | Traumatic brain injury | Brain |
| *van Tilborg et al., 2018* | FA | Immunohistochemistry - MBP | Animal - Rat | White matter injury | Brain |
| *Beckmann et al., 2018* | MTR | Histology - LFB | Animal - Mouse | Demyelination - Cuprizone | Brain |
| *Berman et al., 2018* | MTV | EM - Myelin fraction | Animal - Mouse | Demyelination - Knockout | Brain |
| *Hametner et al., 2018* | R2*, T1, QSM | Histology - LFB | Human | Vascular diseases | Brain |
| *Praet et al., 2018* | MK, RK, AK, FA, MD, RD, AD | Immunohistochemistry - MBP | Animal - Mouse | Amyloidosis | Brain |
| *Wendel et al., 2018* | FA, AD, RD, MD | Immunohistochemistry - MBP | Animal - Mouse | Traumatic brain injury | Brain |
| *West et al., 2018* | MPF, MWF, MVF-T2, MVF-MT | EM - Myelin fraction | Animal - Mouse | Demyelination - Knockout | Brain |
| *Yano et al., 2018* | FA, RD, MD | Immunohistochemistry - PLP | Animal - Mouse | Demyelination - Cuprizone | Brain |
| *Abe et al., 2019* | FA, RD, AD | Microscopy - Myelin thickness | Animal - Mouse | Optogenetic stimulation | Brain |
| *Duhamel et al., 2019* | ihMTR, MTR | Microscopy - Fluorescence | Animal - Mouse | Healthy | Brain |

*Continued on next page*

*Appendix 1—table 1 continued*

| Study | MRI measure(s) | Histology/microscopy measure | Tissue | Condition | Focus |
|---|---|---|---|---|---|
| *Khodanovich et al., 2019* | MPF | Immunohistochemistry - MBP | Animal - Mouse | Demyelination - Cuprizone | Brain |
| *Mollink et al., 2019* | FA | Immunohistochemistry - MBP | Human | Amyotrophic lateral sclerosis | Brain |
| *Peters et al., 2019* | FA, MD | Histology - LFB | Human | Tuberous sclerosis complex | Brain |
| *Pol et al., 2019* | QSM, FA, MD | Histology - Solochrome | Animal - Mouse | Healthy | Brain |
| *Soustelle et al., 2019* | MPF, RD, MWF, rSPF | Immunohistochemistry - MBP | Animal - Mouse | Demyelination - Cuprizone | Brain |
| *Guglielmetti et al., 2020* | MTR, MTR-UTE | Immunohistochemistry - MBP | Animal - Mouse | Healthy | Brain |

