## [Decision Letter]

**Acceptance summary:**

This meta-analysis is a great contribution to efforts in the neuroimaging field to better understand the biological underpinnings of MRI techniques. The work looks at published quantitative relationships between multiple MRI measures that are thought to be sensitive to myelin, and multiple measures from histology. The results are presented in an original, highly interactive manner, which will be a great resource for future studies of myelin in the brain.

**Decision letter after peer review:**

Thank you for submitting your article "An interactive meta-analysis of MRI biomarkers of myelin" for consideration by *eLife*. Your article has been reviewed by three peer reviewers, including Saad Jbabdi as the Reviewing Editor and Reviewer #1, and the evaluation has been overseen by Chris Baker as the Senior Editor. The following individual involved in review of your submission has agreed to reveal their identity: Mark Does (Reviewer #3).

The reviewers have discussed the reviews with one another and the Reviewing Editor has drafted this decision to help you prepare a revised submission.

Below is a compiled summary of all points raised by the three reviewers.

This article presents a meta-analysis of experimental comparisons between MRI and histological measures of myelin. The three reviewers agree that while this is not a particularly novel piece of work, it is well conducted and is presented in a highly original way using interactive visualisation, which is very useful for this type of complex meta analyses.

The reviewers agreed on the following points that need addressing:

1) Relating to previous literature

The primary shortcoming of the manuscript is that while it does a good job of citing prior experimental MRI-histology studies, it does a relatively poor job of providing references for other claims/assertions. We realize that it is not the authors' intention to review the physics or experimental history of myelin imaging methods, but when the authors provide a reference to support a statement, they ought to make it a suitable one (which may require looking more than a couple years back in history). We've listed a few examples below, but the authors should review the entire manuscript with this in mind.

Examples of questionable referencing:

"demyelination is often observed in several neurological diseases such as multiple sclerosis", cite: Wang Y et al., 2015, a paper reporting diffusion spectrum imaging evaluations MS

"measuring myelin in vivo has been an ambitious goal for magnetic resonance imaging (MRI) for almost two decades", cite: Petiet et al., 2019, a review of ultra-high field MRI measures of myelin.

"Diffusion acquisitions are blind to direct myelin measurement (Campbell et al., 2018)"

"A warning message that is evident from these results is the inherent limitation of DWI for estimating myelin content" again, this was studied extensively more than 20 years ago and has been discussed many time since. It's a good to reiterate, but don't make it sound like a novel finding.

"Faster techniques have been proposed for estimating it with gradient- and spin-echo (GRASE) sequences", cite: Faizy et al., 2018. This approach dates back 20 years and was used by Prasloski in 2012 to generate whole cerebrum MWF imaging.

2) Statistical analyses:

– Are the authors able to assess whether the correlations that they report are driven by tissue type differences or finer changes in the degree of myelination?

– Were there interactions between MR technique used and microscopy technique used in the literature? e.g. in Figure 5, are the R^2^ values for "myelin thickness" low because they happened to use diffusion measures rather than MT etc.?

– In general there was not a lot of information on interactions between the variables. Another example: were some techniques more likely to have been done at lower field (there was a strong correlation between field-strength and R^2^)?

– Given that the posed question is "how different are the modalities in their relationship to histology", is there a way to quantify or statistically test the mixed-model findings between modalities to effectively identify if any are better?

– It would be useful to identify the different types of histological techniques alongside the studies for each modality in Figure 4. While this is just one of many factors that is driving the high I^2^, it would allow for the visualisation of the heterogeneity of histological assessments for each modality. Not all histological techniques are born equal and despite the limitations, that the authors have already discussed, electron microscopy might be arguably the best assessment. I suspect due to the high number of MRI modalities and histological techniques and relatively small number of studies, it's not possible to quantify if any modality has a particularly good correlation with any of the two electron microscopy metrics. Still, if possible might be worth doing as EM is the gold-standard for cellular neuroscientists in the myelin field.

3) Overall message:

– Although the authors' discussion and conclusions present a more nuanced view of the findings, this is not clear from the Abstract. At least two MRI markers (MWF and MPF) have fairly high correlations and moderate prediction intervals. We think this could come across a bit more in the Abstract, as it does in their conclusion. Whether these are a true representation of histologically measured myelin is a different question. The authors have discussed this distinction effectively in their manuscript.

– We encourage the authors to consider leaving the reader on a somewhat more positive note. That is, while there is room for more study, this meta-analysis supports the view that myelin imaging with MRI is, in fact, relatively well substantiated by comparisons to histology. In fact, I would say that myelination is one characteristic of tissue that MRI has proven to be able to measure with a good level of specificity!

---

## [Author Response]

This article presents a meta-analysis of experimental comparisons between MRI and histological measures of myelin. The three reviewers agree that while this is not a particularly novel piece of work, it is well conducted and is presented in a highly original way using interactive visualisation, which is very useful for this type of complex meta analyses.

We appreciate the reviewers’ enthusiasm. While we acknowledge that there are already several literature reviews on the topic of MRI-based measures of myelin (as reported in the paper, during the survey we found 50 review articles, with 6 of them specifically discussing comparisons between MRI and histology), to the best of our knowledge no systematic study has been published yet. A search for “myelin meta-analysis” and “myelin systematic review” on PubMed leads respectively to 93 and 91 results (last search: 23/09/20), yet none of these studies focus on MRI validation through histology. We believe that both the systematic and statistical aspects of our work are important, because they give overall quantitative measures of agreement and variability across studies.

The reviewers agreed on the following points that need addressing:1) Relating to previous literatureThe primary shortcoming of the manuscript is that while it does a good job of citing prior experimental MRI-histology studies, it does a relatively poor job of providing references for other claims/assertions. We realize that it is not the authors' intention to review the physics or experimental history of myelin imaging methods, but when the authors provide a reference to support a statement, they ought to make it a suitable one (which may require looking more than a couple years back in history). We've listed a few examples below, but the authors should review the entire manuscript with this in mind.

We thank the reviewers for pointing out this issue in the manuscript. We revised the whole manuscript and provided more suitable references where the previous ones did not properly support our statements (marked through the whole manuscript). We also specifically annotated the manuscript based on the reviewers’ comments 2-6.

Examples of questionable referencing:"demyelination is often observed in several neurological diseases such as multiple sclerosis", cite: Wang Y et al., 2015, a paper reporting diffusion spectrum imaging evaluations MS

We substituted this reference with an updated review on demyelinating diseases from the Clinical Handbook of Neurology (Höftberger and Lassmann, 2018).

"measuring myelin in vivo has been an ambitious goal for magnetic resonance imaging (MRI) for almost two decades", cite: Petiet et al., 2019, a review of ultra-high field MRI measures of myelin.

We substituted this reference with three landmark studies respectively on myelin water imaging, magnetization transfer and T1 mapping (Mackay et al., 1994; Rooney et al., 2007; Stanisz, Kecojevic, Bronskill, and Henkelman, 1999).

"Diffusion acquisitions are blind to direct myelin measurement (Campbell et al., 2018)"

We substituted this reference with two seminal studies that estimated the transverse relaxation times of myelin water molecules and macromolecules (Mackay et al., 1994; Stanisz et al., 1999).

"A warning message that is evident from these results is the inherent limitation of DWI for estimating myelin content" again, this was studied extensively more than 20 years ago and has been discussed many times since. It's a good to reiterate, but don't make it sound like a novel finding.

We agree that this sentence may be misleading, so we clarified it and added relevant references (Beaulieu, 2002, 2009):

“A warning message that is evident from these results is the inherent limitation of DWI for estimating myelin content: this is not by any means a novel result (Beaulieu, 2002, 2009), but it is nevertheless worth reiterating given the outcomes of our analysis.”

We would still like to keep that sentence, as there are still studies using DWI-based measures for estimating myelin content without addressing the related limitations.

"Faster techniques have been proposed for estimating it with gradient- and spin-echo (GRASE) sequences", cite: Faizy et al., 2018. This approach dates back 20 years and was used by Prasloski in 2012 to generate whole cerebrum MWF imaging.

We substituted the previous reference with the original paper presenting the GRASE sequence, its first application to T2 relaxometry and the suggested study by Prasloski and colleagues (Does and Gore, 2000; Feinberg and Oshio, 1991; Prasloski et al., 2012).

2) Statistical analyses:– Are the authors able to assess whether the correlations that they report are driven by tissue type differences or finer changes in the degree of myelination?

As we elaborate more in the next comment, we are not able to quantitatively answer this question given the number of potentially different conditions to consider and the limited number of studies. In more qualitative terms, Figure 6 and the interactive version Figure S7 (https://neurolibre.github.io/myelin-meta-analysis/04/other_factors.html#figure-7) show different correlation ranges depending on the types of tissues considered and on the specific pathology model, but in any case the range observed for white matter, the most common tissue studied, is particularly large, suggesting that tissue type differences are not the main factor affecting correlation. We added this consideration in the text:

“The effect of considering different types of tissues is showed in Figure 6 and Figure S7, where correlation ranges change when considering different types of tissue. However, the large correlation range in white matter, the most common tissue studied, suggests that other factors also affect the correlation.”

– Were there interactions between MR technique used and microscopy technique used in the literature? E.g. in Figure 5, are the R^2^ values for "myelin thickness" low because they happened to use diffusion measures rather than MT etc.?

As noted by the reviewers in this and the following comment, interactions are key elements in this kind of analysis. Unfortunately, given the limited number of studies we are not able to fit a model to study those interactions: specifically, we do not have enough studies to represent each possible MRI/microscopy combination. Despite not being able to tackle this question quantitatively, we believe that the provided interactive visualization is an effective way to qualitatively explore this kind of questions. We added these considerations:

“Given the limited number of studies, it is not possible to quantitatively study interactions between MRI measures and the other factors (e.g. modality used as a reference, tissue types, magnetic field strength). For further qualitative insights, we invite the reader to explore the interactive Figures S7-S8. A first important factor to consider is the validation modality used as a reference, which will be dictated by the equipment availability and cost.”

Regarding the specific issue raised by the reviewers, Figure S7 (https://neurolibre.github.io/myelin-meta-analysis/04/other_factors.html#figure-7) allows the reader to hover over each point in the top plot and get a sense of which MRI measures were investigated for each microscopy technique (screenshot attached). Myelin thickness was used as reference for twelve measures that included diffusion, relaxometry and magnetization transfer.

We believe that having access to the interactive figures directly in an executable research article on the Stencila platform will allow a more immediate exploration of our results.

– In general there was not a lot of information on interactions between the variables. Another example: were some techniques more likely to have been done at lower field (there was a strong correlation between field-strength and R^2^)?

Following up on the previous comment, Figure S8 (https://neurolibre.github.io/myelin-meta-analysis/04/other_factors.html#figure-8) shows the MRI measures as a function of the related magnetic field strengths, using the same colour coding as Figure 2. One can notice how most studies have been done at 7T and 9.4T, while the first studies (chronologically) were performed at 1.5T. Few measures were studied at other field strengths. We added this consideration in the text:

“A further example of influential factor often dictated by equipment availability is the magnetic field strength of the MRI scanner: Figure S8 shows that most studies were conducted at 7T and 9.4T, with some pioneering studies at 1.5T and even fewer ones at other field strengths.”

– Given that the posed question is "how different are the modalities in their relationship to histology", is there a way to quantify or statistically test the mixed-model findings between modalities to effectively identify if any are better?

Following the reviewers’ suggestion, we performed an additional analysis using a repeated measures meta-regression, explained in the text:

“For the explicit purpose of comparing the effect sizes between different MRI measures, we conducted a repeated measures meta-regression on every R^2^ value recorded. […] While the repeated measures meta-regression makes direct comparisons straightforward, we reported the main R^2^ estimates based on the measure-specific mixed-effects models, as they make weaker assumptions.”

From this analysis, we observed both significant differences and comparable R^2^ estimates, overall subdividing the MRI measures in two groups, with magnetization- and relaxometry-based ones providing higher estimates and diffusion-based measures providing lower estimates. The new results are reported in the text and discussed in subsection “Meta-analysis”:

“To investigate the significance of the differences between measures, we conducted a repeated measures meta-regression on every R^2^ estimate recorded (98 in total over 43 studies). […] From this perspective, MPF has higher R^2^ estimates compared to all the other measures, but it is only marginally higher than MWF (z-score=0.77; p-value=1) so we cannot claim that one is superior to the other. Following the same reasoning, MTR and T1 are not statistically different (z-score=0.47; p-value=1).”

The repeated measure meta-regression confirms this overall picture, clearly distinguishing MT- and relaxometry-based measures from diffusion-based ones (Figure 5).

– It would be useful to identify the different types of histological techniques alongside the studies for each modality in Figure 4. While this is just one of many factors that is driving the high I^2^, it would allow for the visualisation of the heterogeneity of histological assessments for each modality. Not all histological techniques are born equal and despite the limitations, that the authors have already discussed, electron microscopy might be arguably the best assessment. I suspect due to the high number of MRI modalities and histological techniques and relatively small number of studies, it's not possible to quantify if any modality has a particularly good correlation with any of the two electron microscopy metrics. Still, if possible might be worth doing as EM is the gold-standard for cellular neuroscientists in the myelin field.

We agree that it is useful to have a sense of which histological techniques were used for each MRI measure: we added this information in Figure S5 (https://neurolibre.github.io/myelin-meta-analysis/03/meta_analysis.html#figure-5), when hovering on each point (screenshot attached). Although this information is currently not visible in the static figure in the manuscript, it will be immediately accessible in Stencila.

As mentioned in comment 8 and as the reviewers already acknowledge, unfortunately the number of studies using EM is not sufficient to make more definitive statements for each MRI modality.

3) Overall message:– Although the authors' discussion and conclusions present a more nuanced view of the findings, this is not clear from the Abstract. At least two MRI markers (MWF and MPF) have fairly high correlations and moderate prediction intervals. We think this could come across a bit more in the Abstract, as it does in their conclusion. Whether these are a true representation of histologically measured myelin is a different question. The authors have discussed this distinction effectively in their manuscript.

We agree that the Abstract was not able to present the overall picture as in the discussion. We rephrased part of the Abstract taking into account both this comment and the journal’s word count limit of 150 words:

“We report the overall estimates and the prediction intervals for the coefficient of determination and find that MT and relaxometry-based measures exhibit the highest correlations with myelin content. We also show which measures are, and which measures are not statistically different regarding their relationship with histology.”

– We encourage the authors to consider leaving the reader on a somewhat more positive note. That is, while there is room for more study, this meta-analysis supports the view that myelin imaging with MRI is, in fact, relatively well substantiated by comparisons to histology. In fact, I would say that myelination is one characteristic of tissue that MRI has proven to be able to measure with a good level of specificity!

We added a brief paragraph towards the end of the Discussion to emphasize the positive message of our results:

“We hope this meta-analysis convinces the reader that a holy grail of myelin imaging does not exist, at least as long as we consider histology to be the ground truth. Given that we all have to pick our poison, the upside is that measures based on MT and relaxometry are not statistically different, and therefore future studies have an actual choice among candidate measures.”